# Myocardial Perfusion Single-Photon Emission Computed Tomography (SPECT) Image Denoising: A Comparative Study

**DOI:** 10.3390/diagnostics13040611

**Published:** 2023-02-07

**Authors:** Abdurrahim Rahimian, Mahanaz Etehadtavakol, Masoud Moslehi, Eddie Y. K. Ng

**Affiliations:** 1Department of Medical Physics, School of Medicine, Isfahan University of Medical Sciences, Isfahan 81745-33871, Iran; 2School of Mechanical and Aerospace Engineering, College of Engineering, Nanyang Technological University, Singapore 639798, Singapore

**Keywords:** single-photon emission computed tomography, myocardial perfusion, denoising, filter

## Abstract

The present study aimed to evaluate the effectiveness of different filters in improving the quality of myocardial perfusion single-photon emission computed tomography (SPECT) images. Data were collected using the Siemens Symbia T2 dual-head SPECT/Computed tomography (CT) scanner. Our dataset included more than 900 images from 30 patients. The quality of the SPECT was evaluated after applying filters such as the Butterworth, Hamming, Gaussian, Wiener, and median–modified Wiener filters with different kernel sizes, by calculating indicators such as the signal-to-noise ratio (SNR), peak signal-to-noise ratio (PSNR), and contrast-to-noise ratio (CNR). SNR and CNR were highest with the Wiener filter with a kernel size of 5 × 5. Additionally, the Gaussian filter achieved the highest PSNR. The results revealed that the Wiener filter, with a kernel size of 5 × 5, outperformed the other filters for denoising images of our dataset. The novelty of this study includes comparison of different filters to improve the quality of myocardial perfusion SPECT. As far as we know, this is the first study to compare the mentioned filters on myocardial perfusion SPECT images, using our datasets with specific noise structures and mentioning all the elements necessary for its presentation within one document.

## 1. Introduction

It is estimated that 17.9 million deaths are caused each year by the cardiovascular diseases, including coronary artery diseases (CAD) such as angina and myocardial infarction [1]. Non-invasive myocardial perfusion imaging (MPI) plays an important role in the diagnosis of CAD [2]. By MPI tests, regional changes in blood flow patterns can be assessed during various physiological conditions of rest and cardiovascular stress (e.g., normal vs. ischemic and fibrotic scar) [3]. Additionally, it provides information on the myocardial function that is superior to anatomic assessment alone [4]. MPI can be completed with positron emission tomography (PET), single-photon emission computed tomography (SPECT), magnetic resonance imaging (MRI), or computed tomography (CT) tests. Among the mentioned non-invasive methods, PET is considered the gold standard. However, it is not widely used since it is expensive in terms of purchasing equipment compared to PET; SPECT-MPI is cheaper and more readily available. SPECT-MPI is widely used to diagnose patients with or suspected of having CAD. It can provide important information about myocardial perfusion and function [5]. SPECT images are commonly noisy, because gamma rays are attenuated and scattered before they hit the detector [6]. In digital image processing, finding a technique that can remove noise from SPECT images has been a long-term issue, because these images have high noise level and low contrast.

Noise in nuclear medicine images is identified as random and structural noises. Random noise results from statistical variations in the count rate, which is related to the information density measured as counts per unit area. While structural noise can be described as non-random asymmetries in the distribution of radioactivity, which reduces the structural information of the target organ [7]. The main noise sources in SPECT images include inherent random changes in the unlimited photon count, electronic tracking, and data recording systems, and limited time for preparing each slice [8,9]. Therefore, investigating the types of noises and their sources requires designing a noise-removal algorithm for SPECT images. Usually, SPECT images contain noise with an independent and uniform distribution, which has a Gaussian nature throughout the image [8]. Filtering is a technique used to modify or improve image quality. For instance, an image can be filtered to emphasize some features or remove other features. The image processing operations involved in filtering include smoothing, sharpening, and edge enhancement. Filtering is a neighborhood operation in which the value of any given pixel in the output image is determined by applying an algorithm to the values of the neighboring pixels of the corresponding input pixel. Occasionally, the low-frequency signals of noisy images must be preserved, and high-frequency signals are removed for optimal filtering. Consequently, a low-pass filter is appropriate for this task [10]. 

Masoomi et al. (2019) [11] conducted phantom chest imaging to re-evaluate 92 male and female patients (41–77 years) with suspected CAD. They determined that among the Hanning, Hamming, Parzen, Butterworth, Metz, and Wiener filters, the Wiener filter produced the highest contrast results for cold and warm spheres in the phantom and for the patient. Sayed and Ismail (2020) [12] investigated the effects of two types of reconstruction filters, namely Butterworth and Hamming, on the quality of images in hot and cold regions. In their study, the Butterworth filter showed more hot and cold regions in the reconstructed images. It also had higher contrast values than those of the Hamming filter. However, with the Butterworth filter, a reduction in the signal-to-noise ratio (SNR) was achieved for both regions, as the cut-off frequency increased relative to the Hamming filter. Kim et al. (2021) [13] investigated the usefulness of the MMWF algorithm for improving SPECT images using a three-dimensional Hoffmann brain phantom injected with radioisotope T 99mc . They showed that brain images were improved using the MMWF algorithm. A more comprehensive comparison of the literature is presented in Table 1.

Despite these efforts, limited studies have been accomplished to assess the application of the modified–median Wiener (MMWF) filter to myocardial perfusion SPECT images. Therefore, the present study aimed to evaluate the effectiveness of using these filters to increase the quality of myocardial perfusion SPECT images by comparing their signal-to-noise ratio (SNR), peak signal-to-noise ratio (PSNR), and contrast-to-noise ratio (CNR).

Subsequently, materials and methods are explained in Section 2. Section 3 presents the results and discussion, and Section 4 presents the conclusion.

## 2. Materials and Methods

### 2.1. Materials

#### Image Preparation and Data Collection

Nuclear cardiology is expanding within nuclear medicine applications, even today. Identifying each patient’s characteristics and conditions permits nuclear medicine technologists to adjust the protocol, resulting in significant advantages for every patient. In this regard, the Technologist Committee of the European Association of Nuclear Medicine (EANM) proposed recommendations for best practice in computed tomography applied to attenuation correction and calcium score in myocardial perfusion imaging, which defines the set of knowledge, skills, and competencies for nuclear medicine technologists [15]. 

Our dataset was provided according to EANM procedural guidelines for radionuclide myocardial perfusion imaging with SPECT and SPECT/CT [16]. Thirty-two patients, including 17 women and 15 men, were considered. Two women were excluded from our dataset due to pregnancy. Accordingly, myocardial perfusion SPECT images of 30 patients (15 men and 15 women) aged 47–75 years with a body mass index of 28.3 ± 4.1 were included. These images were obtained at the Shahid Chamran Hospital of Isfahan, Iran, using a Siemens Symbia T2 dual-head SPECT/CT scanner. Imaging was performed in two phases: rest and stress tests. In the rest test, based on the patient’s weight, 15–20 mCi T 99mc−MIBI  (99m Tc-methoxy isobutyl isonitrile) was injected intravenously into the patient, and imaging was repeated after 45 min. In the stress test, the patient’s heart rate was increased by exercise (treadmill) or by administering special drugs, such as dipyridamole, and then 20–25 mCi T 99mc−MIBI was injected into the patient. Subsequently, another image was captured after 15–45 min. Table 2 lists the specifications of the imaging performed in this study. The T 99mc−MIBI, or T 99mc-sestamibi, is used for diagnostic purposes in cardiac, breast, and parathyroid tissues. As T 99mc−MIBI is released slowly from myocardial tissue after cellular uptake, higher-quality imaging is facilitated [17]. 

### 2.2. Methods

The steps of this study are as follows: first, data were collected at Chamran Hospital, Iran. Second, filters such as Butterworth, Hamming, Gaussian, and Wiener, with different kernel sizes, and MMWF with different kernel sizes were applied to enhance the quality of myocardial perfusion SPECT images. Third, the qualities of the SPECT images were evaluated by calculating indicators, such as SNR, PSNR, and CNR. Moreover, statistical analysis was performed using one way-ANOVA test. Finally, the obtained indicators were compared and the best filter for denoising our dataset was selected and introduced. These steps are demonstrated in Figure 1.

#### 2.2.1. Filtering

Filters such as Butterworth, Hamming Gaussian, Wiener, and MMWF are often used for SPECT image denoising, to limit the effect of noise on the interpretation and analysis of images [18]. These filters can be applied in both spatial and frequency domains. They were used in this study, which is briefly explained below.

(1)Butterworth filter

The Butterworth filter is the most common low-pass filter in nuclear medicine and is characterized by two parameters: (1) cut-off frequency, which is the frequency at which the filter function goes to zero; and (2) order, which changes the slope of the filter [19]. Regarding the ability to change the cut-off frequency and the filter’s slope, the Butterworth filter can reduce noise and maintain image clarity. The Butterworth filter in the frequency domain is defined as:(1)$$B\left(f\right)=\frac{1}{\left[1+{\left(\frac{f}{f_{c}}\right)}^{2n}\right]}\;\;\;\;\;$$
where f is the spatial frequency domain, $f_{c}$ is the cut-off frequency, and *n* is the filter order.

As shown in Figure 2, as *n* approaches infinity, the Butterworth function becomes a rectangular function, and frequencies lower than $f_{c}$ are transferred while frequencies higher than $f_{c}$ are removed. The larger *n* becomes, the sharper the filter function, and vice versa. In addition, with increasing the cut-off frequency increases, more frequencies are transmitted. In this study, we investigated *n* levels ranging from 5 to 10 and cut-off frequencies from 0.3 Nq (Nyquist frequency) to 0.7 Nq.

(2)Gaussian filter

This filter, which is used to remove noise or high-frequency components of an image, is defined as follows [20]:(2)$$G\left(u,v\right)=e^{-\frac{D{\left(u,v\right)}^{2}}{2\sigma^{2}}}\;\;\;\;$$
where D(u,v) is the distance between points (u,v) to the center of the filter and σ is the standard deviation. When applied in two dimensions, this formula produces a surface whose lines are concentric circles with a Gaussian distribution from the center point. As shown in Figure 3, with an increase in *σ*, the slope of the graph decreases, and more frequencies are transmitted.

(3)Wiener filter

By considering the characteristics of statistical noise in a degraded image, the Wiener filter reduces the noise distribution in the frequency domain and works based on the principle of local image variance calculation. Therefore, if the local variance of the image is high, denoising is performed poorly; when the local variance is small, a more accurate image can be obtained. However, it requires more computing time [21]. The filter is defined as follows:(3)$$\mu =\frac{1}{xy}\;\underset{x,y\in \eta }{\sum }a\left(x,\;y\right)\;$$
(4)$$\sigma^{2}=\frac{1}{xy}\;\underset{x,y\in \eta }{\sum }a{\left(x,y\right)}^{2}-\mu^{2}$$
(5)$$F_{W}\left(x,y\right)=\mu +\frac{\sigma^{2}+\nu^{2}}{\sigma^{2}}\;.\;\left(a\left(x,y\right)-\mu \right)\;$$
where *μ* is the mean value of the pixel, $\;\sigma^{2}$ is the variance of the Gaussian noise in the image, and x×y is the size of the neighborhood area in the kernel size *η*. $\nu^{2}$ is the noise variance in kernel size. If the noise variance is not provided as an input, the average of all local variances estimated for each kernel is used [22]. In this study, we used the second variance adjustment.

(4)MMWF filter

Cannistraci et al. [22] designed the MMWF filter to reduce noise distribution in degraded images. The MMWF filter is beneficial, because it can be easily used in the spatial domain. Moreover, edges are better preserved in an image than those in other common noise reduction filters, such as averaging filters. The MMWF, a filter based on the Wiener filter, replaces the pixel values of the kernel matrix with median values, thereby reducing noise in the degraded image [23,24]. The MMWF filter is defined as follows:(6)$$F_{MMWF}\left(x,y\right)=\tilde{\mu }+\frac{\sigma^{2}+\nu^{2}}{\sigma^{2}}\;.\;\left(a\left(x,y\right)-\tilde{\mu }\right)\;$$
where $\tilde{\mu }$ is the median value of kernel sizes.

(5)Hamming filter

The Hamming filter is a low-pass filter with one variable parameter called the cut-off frequency. It is defined as follows [25]:(7)$$H\left(f\right)=\{\begin{array}{c} 0.54+0.46\cos\left(\frac{\pi f}{f_{m}}\right)\;\;\;\left(0\;\leq \left|f\right|\leq f_{m}\right)\\ 0\;\;\;\;\;\;\;\;\;\;\;\;\;\;\;\;\;\;\;\;\;\;\;\;\;\;\;\;\;\;others \end{array}$$
where *f* is the spatial frequency of the image and *f_m_* is the cut-off frequency. In this study, the cut-off frequencies of 0.3 Nq to 0.7 Nq was investigated.

#### 2.2.2. Quantitative Indicators Calculations

Several quantitative indices, such as SNR, CNR, and PSNR, are available to evaluate the quality of the filtered images. The higher the SNR value, the stronger the signal in relation to the noise levels will be. Conversely, more useful information is received as a signal than unwanted information or noise. CNR is similar to the SNR used to determine the image quality; however, it subtracts one term before obtaining the ratio. This issue is important when there is a significant bias in an image, such as fog. Thus, an image may have a high SNR but a low CNR.

This study used these three indices to evaluate the quality of SPECT images. $\;ROI_{A}\;\;$(Regions of interest), or target regions, and $ROI_{B}\;$, or background regions, were selected, as shown in Figure 4 for a SPECT image. The SNR and CNR values were obtained as follows:(8)$$SNR=\frac{S_{A}}{\sigma_{A}}$$
(9)$$CNR=\frac{\left|S_{A}-S_{B}\right|}{\sqrt{\sigma_{A}+\sigma_{B}}}$$
where $S_{A}$ and $\sigma_{A}$ represent the mean and standard deviation for $ROI_{A}$, respectively. Additionally, $S_{B}$ and $\sigma_{B}$ represent the mean and standard deviation for $ROI_{B}$, respectively. Moreover, the size of $ROI_{A}$ and $ROI_{B}$ is equal to 10 × 10.

The PSNR is a measure of the ratio of the maximum possible value of the signal to the power of noise distortion, which affects the display quality. Because many signals have a wide dynamic range (the ratio of the largest and smallest possible values of a variable quantity), the PSNR is usually expressed in a logarithmic decibel scale. The PSNR equation is as follows [26]:(10)$$PSNR=10\log_{10}\left(\frac{I_{max}{}^{2}}{MSE\left(G,X\right)}\right)\;$$

The mean square error (MSE) is defined as follows:(11)$$MSE=\frac{\sum_{i=1}^{M}\sum_{j=1}^{N}{\left(G_{ij}-X_{ij}\right)}^{2}}{3\;MN}$$

*M* and *N* represent the size of the image matrix, and *G* and *X* represent the filtered and unfiltered images, respectively. The novelty of this study lies in the following: (1) comparison of different filters to improve the quality of myocardial perfusion SPECT images and (2) using our datasets with specific noise structures.

#### 2.2.3. Statistical Analysis

Analysis of PSNR, SNR, and CNR was achieved using IBM SPSS software, version 26. It is worth nothing that one way-ANOVA test was used in this study, with *p* < 0.05.

## 3. Results

The average indices of SNR, PSNR, and CNR for myocardial perfusion SPECT images of 30 patients with the Hamming filter and the cut-off frequencies of 0.3 Nq to 0.7 Nq are demonstrated in Figure 5. Moreover, they are presented in Figure 6 for the Butterworth filter of order 5 to 10 and cut-off frequencies of 0.3 Nq to 0.7 Nq. 

Figure 7 shows the original and denoised images using Butterworth (*n* = 5, *fc* = 0.3 Nq), Gaussian, Wiener (3 × 3), MMWF (3 × 3), Wiener (5 × 5), MMWF (5 × 5), and Hamming filters.

Figure 8, Figure 9 and Figure 10 show the graphs of the average SNR, PSNR, and CNR obtained by Butterworth (*n* = 5, *fc* = 0.3 Nq), Gaussian, Wiener (3 × 3), MMWF (3 × 3), Wiener (5 × 5), MMWF (5 × 5), and Hamming filters, respectively. Additionally, their numerical values can be seen and compared in Table 3, and the statistical analysis results using one way-ANOVA are demonstrated in Table 4. 

## 4. Discussion

The parameters that are effective in choosing the type of filter for SPECT images are: (1) isotope energy, number of counts per unit area; (2) amount of statistical noise and background noise; (3) type of organ to be imaged; (4) type of information required from the images; and (5) the collimator used in imaging [27,28]. 

Park et al. (2020) [14], in a study based on the NEMA IEC body phantom, showed that the MMWF filter outperforms the Wiener, Gaussian, and median filters in reducing noise distribution in gamma camera images. The image quality improved from 20.6% to 65.5%, 7.4% to 40.3%, and 12.7% to 44.7% for the SNR, COV (Coefficients of variation), and CNR values, respectively, when using the MMWF filter. Since the parameters chosen in our study were not the same as those used in Park et al.’s study, our results were not the same as theirs. Notably, we used myocardial perfusion SPECT images, whereas Park et al. used the NEMA IEC body phantom image. In addition, in their study, the activity concentration ratio of T 99mc in the $ROI_{A}$ to $ROI_{B}$ was 8:1. However, calculation of this ratio was one of the limitations of our study. Using a gamma camera, these authors determined the image recording time to be 50 min. In contrast, in our study, each image took 20 s as it was registered.

According to Figure 5, the Hamming filter with a cut-off frequency of 0.3 Nq obtained the highest SNR and CNR indices of 4.16 and 2.41, respectively. The highest PSNR index of 34.49 (dB) was achieved with a cut-off frequency of 0.7 Nq. As we can see from Figure 6, the Butterworth filter of 5th order and a cut-off frequency of 0.3 Nq obtained the highest SNR and CNR indices of 4.34 and 2.35, respectively. Additionally, the highest value of the PSNR index corresponded to the 10th order and a cut-off frequency of 0.7 Nq was 42.46 (dB). As shown in Figure 7, the Wiener filter (5 × 5) does not preserve the edges as clearly as the MMWF (5 × 5) filter does; however, it outperformed the Butterworth, Gaussian, MMWF (3 × 3), MMWF (5 × 5), and Hamming filters, with greater SNR and CNR indices improving the qualities of our SPECT dataset.

According to Table 3 and Table 4, the Wiener (5 × 5) filter has the highest value of SNR index, which was equal to 4.90 ± 0.69. In addition, it had statistically significant differences with other filters (*p* < 0.05). In this index, Butterworth, Wiener (3 × 3), and MMWF (3 × 3) filters had no significant differences (*p* > 0.05). In addition, Table 3 shows that the highest value of the PSNR index corresponds to the Gaussian filter with a value of 50.09 ± 10.63 (dB). However, this filter was significantly different from other filters (*p* < 0.05). It is worth noting that the lowest value of PSNR index, with the value of 29.93 ± 3.13 (dB), corresponds to the Wiener (5 × 5) filter. In this index, there was no significant difference between the Wiener (5 × 5) and MMWF (5 × 5) (*p* = 0.281) filters, or between the Wiener (3 × 3) and MMWF (3 × 3) (*p* = 0.143). Additionally, the highest value of the CNR index was related to the Wiener (5 × 5), with a value of 2.65 ± 0.57. Significant differences were observed between the Wiener (5 × 5) and other filters (*p* < 0.05), but no significant differences were observed between Butterworth, Wiener (3 × 3) and MMWF (3 × 3) filters, or between Gaussian and MMWF (3 × 3) filters (*p* > 0.05). For SNR and CNR indices, there was no significant difference between the Hamming filter and the MMWF (3 × 3). However, the MMWF (3 × 3) filter not only preserved the edge of the image better than the Hamming filter, but also obtained a higher value in the PSNR index. Therefore, it can be concluded that the MMWF (3 × 3) filter performed better than the Hamming filter when denoising myocardial perfusion SPECT images. According to Table 3, the Butterworth filter achieved higher SNR and PSNR values than the Hamming filter, which was statistically significant. Although the Hamming filter obtained a higher CNR index than the Butterworth filter, there was no statistically significant difference (*p* = 0.329). Therefore, the Butterworth filter performed better than the Hamming filter, as concluded by the study conducted by Sayed and Ismail (2020) [12].

## 5. Conclusions

The results show that the Wiener filter, with a kernel size of 5 × 5, outperformed other filters like Butterworth, Hamming, Gaussian, and Wiener, with a kernel size of 3 × 3, and MMWF, with kernel sizes of 3 × 3 and 5 × 5, when denoising the myocardial perfusion SPECT images with T 99mc−MIBI that were investigated in this study.

## Figures and Tables

**Figure 1 diagnostics-13-00611-f001:**
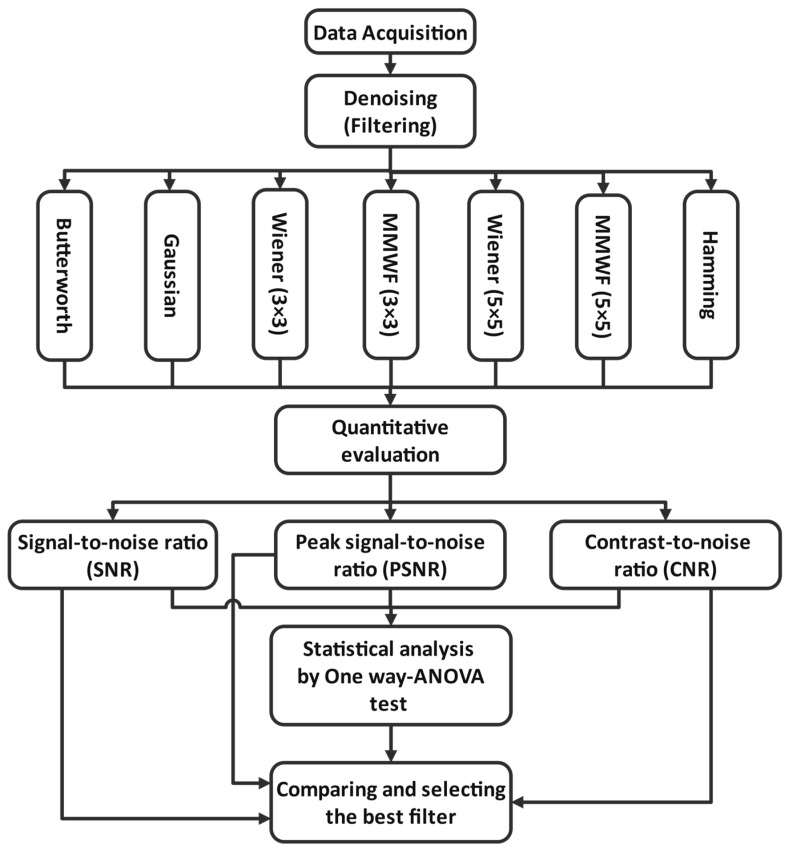
Steps followed in this study (x × x represents the kernel size).

**Figure 2 diagnostics-13-00611-f002:**
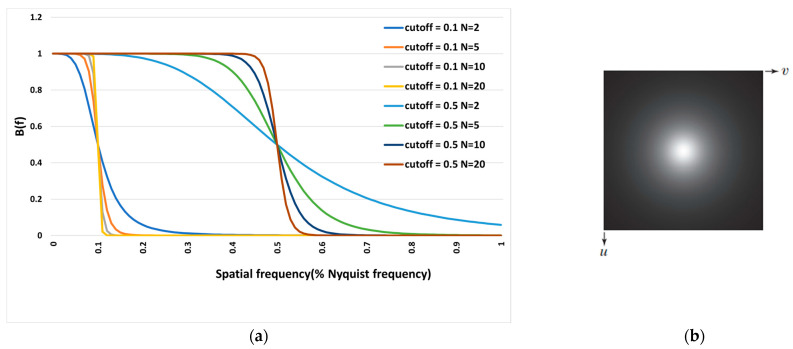
Diagram (**a**) of the Butterworth filter in terms of spatial frequency (Nyquist frequency percentage) and (**b**) the representation of the Butterworth filter function as an image.

**Figure 3 diagnostics-13-00611-f003:**
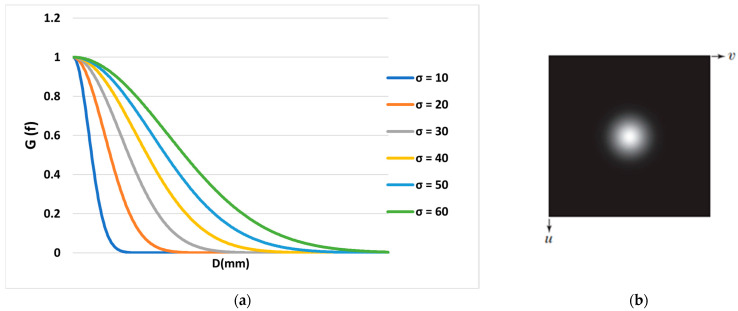
(**a**) Gaussian filter diagram according to the distance from the point to the center of the filter and (**b**) Gaussian filter function display as an image.

**Figure 4 diagnostics-13-00611-f004:**
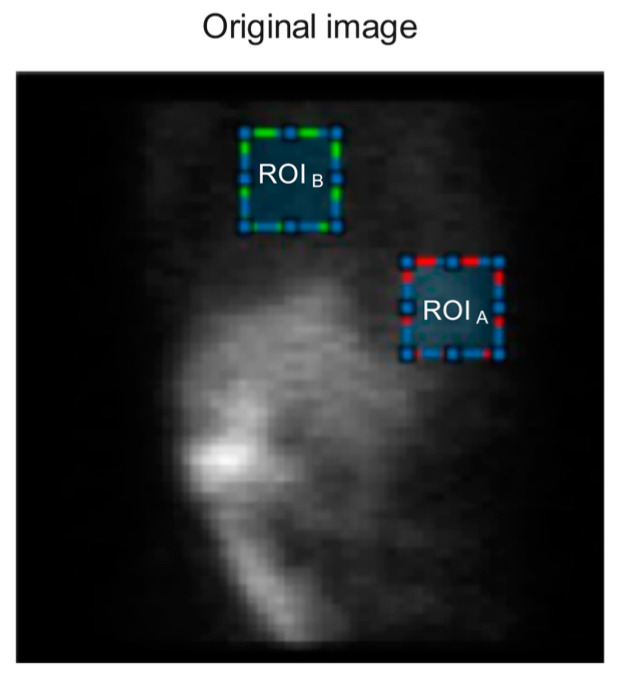
A SPECT image showing region of interest A $\left(ROI_{A}\right)$ with a highlighted red square, as well as the background region ($ROI_{B})$ with a highlighted green square, for calculation of the signal-to-noise ratio (SNR) and contrast-to-noise ratio (CNR) indicators.

**Figure 5 diagnostics-13-00611-f005:**
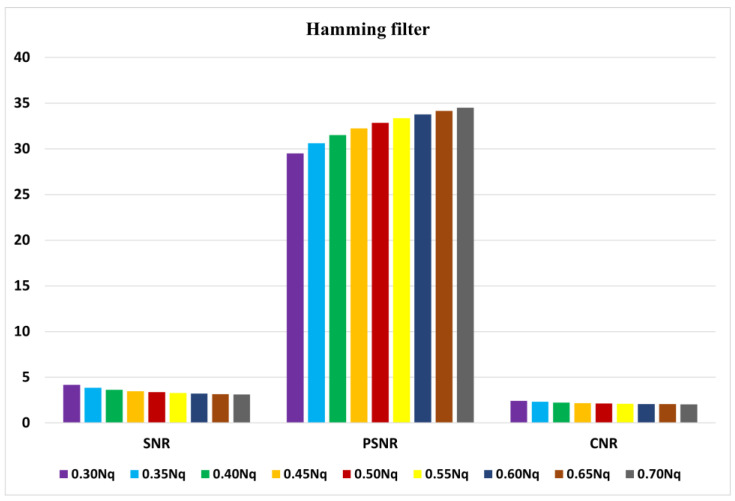
Graph of average of signal-to-noise ratio (SNR), peak signal-to-noise ratio (PSNR), and contrast-to-noise ratio (CNR) using the Hamming filter.

**Figure 6 diagnostics-13-00611-f006:**
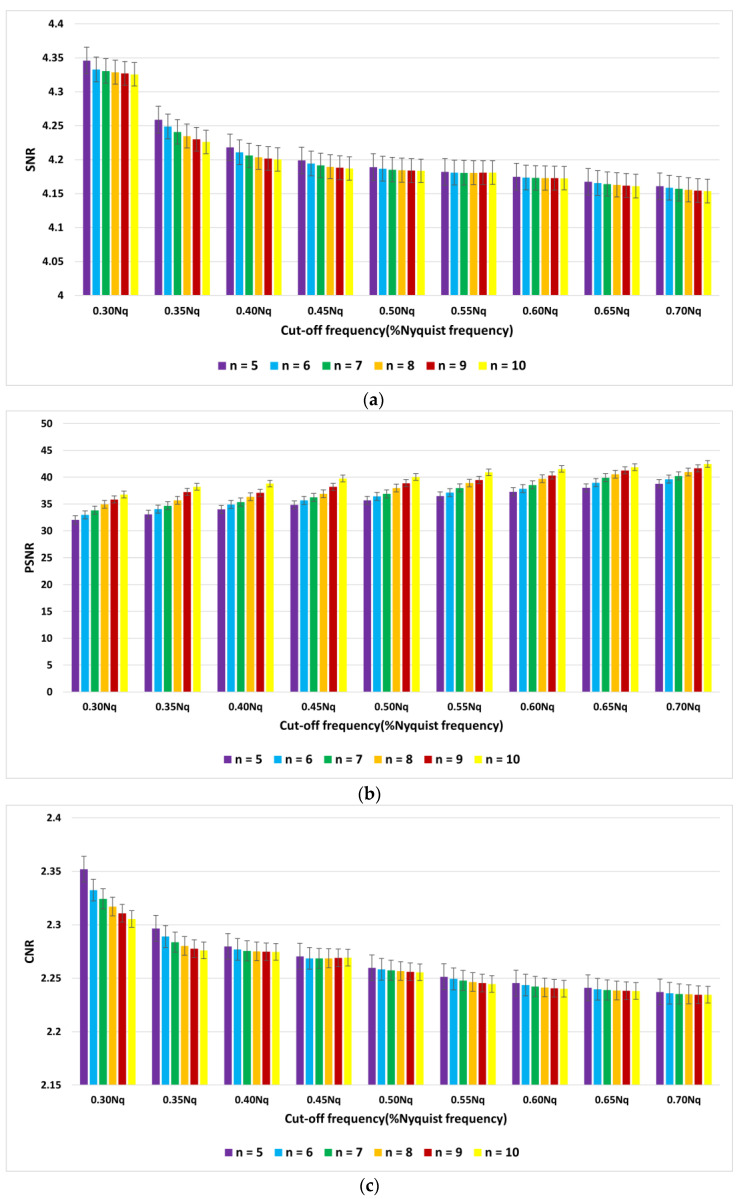
Graph of average of (**a**) signal-to-noise ratio (SNR), (**b**) peak signal-to-noise ratio (PSNR), and (**c**) contrast-to-noise ratio (CNR) using the Butterworth filter.

**Figure 7 diagnostics-13-00611-f007:**
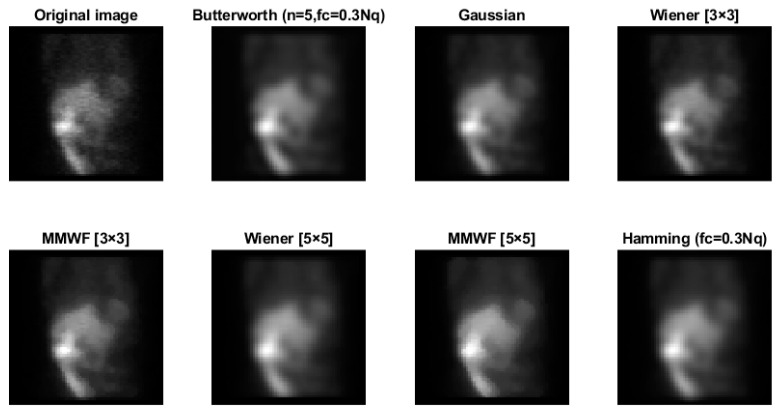
Images obtained by filtering the image of Figure 3 with Butterworth (*n* = 5, fc = 0.3 Nq), Gaussian, Wiener (3 × 3), MMWF (3 × 3), Wiener (5 × 5), MMWF (5 × 5), and Hamming (fc = 0.3 Nq) filters (x × x represents the kernel size).

**Figure 8 diagnostics-13-00611-f008:**
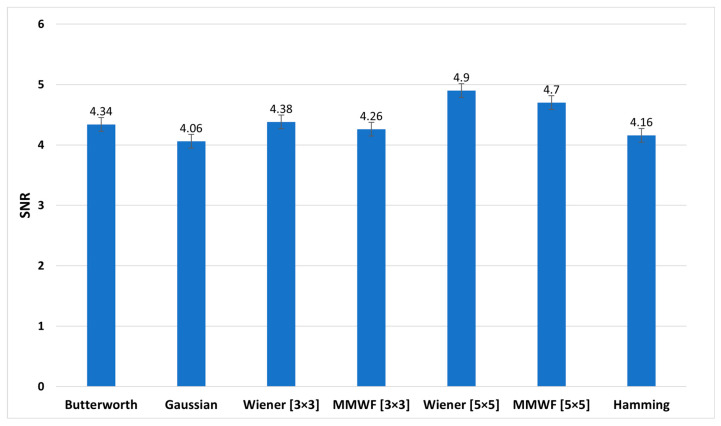
Comparison of the average signal-to-noise (SNR) graph of different filters for the studied myocardial perfusion SPECT images.

**Figure 9 diagnostics-13-00611-f009:**
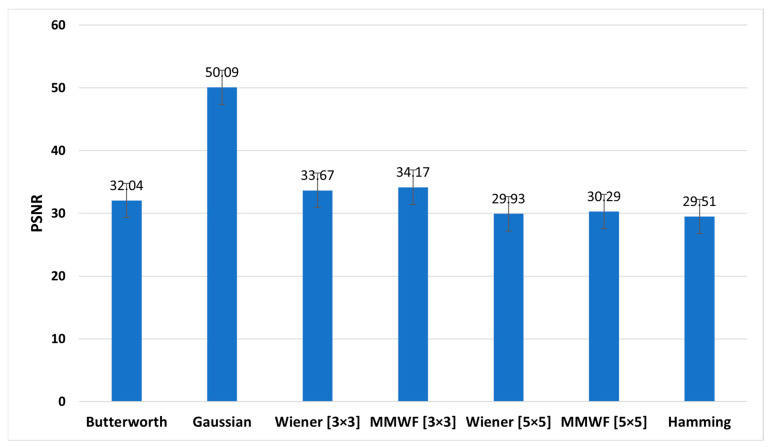
Comparison of the average peak signal-to-noise (PSNR) graph of different filters for the studied myocardial perfusion SPECT images.

**Figure 10 diagnostics-13-00611-f010:**
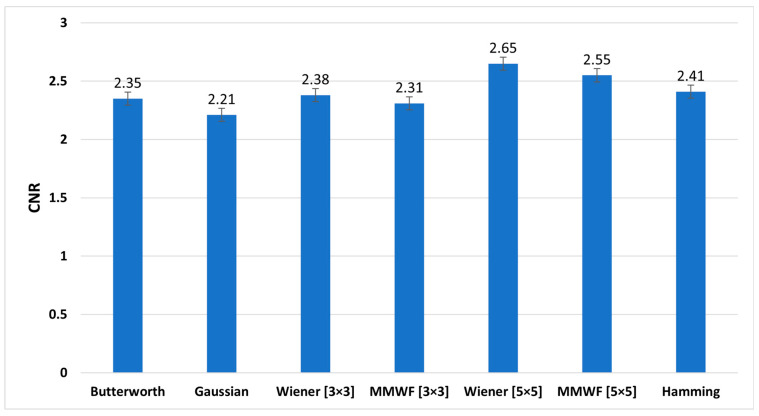
Comparison of average contrast-to-noise (CNR) graph of different filters for studied myocardial perfusion SPECT images.

**Table 1 diagnostics-13-00611-t001:** Literature table.

Authors	Materials and Methods	Results	Conclusion
Masoomi et al. (2019) [11]	Data: images of chest phantoms as well as 92 patients (41–77 years) with CAD filters: Butterworth, Metz, Hamming, and Wiener Radioisotope: T 99mc Reconstruction method: filtered back projection (FBP)	The Wiener, Metz, and Butterworth filters provided the highest contrast (99–66.4%) and (81–32%) for the cold and hot inserts, respectively. Additionally, they reported that patients’ scans, which was denoised with the Wiener filter, presented an elevated diagnostic accuracy and correlated well with the CT-angio and angiography results (*p* < 0.001 and r = 0.79 for Wiener and *p* = 0.004 and r = 0.39 for Butterworth).	The Wiener filter could present results with the highest contrast for phantom imaging of various cold and hot spheres and for the patient images that were more consistent with angiography results
Sayed and Ismail (2020) [12]	Data: acrylic cylindrical phantom image Filters: Butterworth, and Hamming Radioisotope: T 99mc Reconstruction method: FBP	The Butterworth filter was able to display more hot and cold regions in the reconstructed images and obtained higher contrast values than the Hamming filter. Additionally, the Butterworth filter increased the SNR for both regions and decreased the cut-off frequency compared to the Hamming filter	The Butterworth filter provided better results than the Hamming filter. Both filters had different effects on the quality of hot and cold regions with changing the cut-off frequency
Park et al. (2020) [14]	Data: the NEMA IEC body phantom image Filters: median, Gaussian, Wiener, and median modified Wiener filter (MMWF) Radioisotope: T 99mc Reconstruction method: not defined	Respectively, the MMWF, Median, Gaussian, and Wiener filters were more effective in improving image quality. Additionally, the results showed that SNR, COV, and CNR values were improved from 20.6 to 65.5%, 7.4 to 40.3%, and 12.7 to 44.7%, respectively, using the MMWF filter	The MMWF filter is more helpful for reducing the noise distribution in gamma camera images than the median, Gaussian, and Wiener filters
Kim et al. (2021) [13]	Data: the 3D Hoffman brain phantom image Filters: MMWF Radioisotope: T 99mc Reconstruction methods: FBP and ordered subset expectation maximization (OSEM)	The SNR and CNR with the OSEM reconstruction method were 37.7% and 25.9% higher than the FBP reconstruction method, respectively. In addition, using the MMWF filter, the average SNR and CNR values were 35.9% and 17.1% higher than the values without the MMWF filter	The MMWF filter, regardless of what reconstruction method was used, improved image quality in SPECT images, and the OSEM reconstruction method was more efficient than the FBP reconstruction method

**Table 2 diagnostics-13-00611-t002:** Characteristics of myocardial perfusion SPECT images.

Parameter	Amount/Attribute	Parameter	Amount/Attribute
Radioisotope	T 99mc−MIBI	Degrees of rotation	90
Matrix size	64 × 64	Number of views	16
Collimator	Low energy/high resolution	The recording time of each view	20 s
Zoom	1.45	Detector configuration	90
Patient position	Supine	Mode	Step and shoot
Rotate direction	Counterclockwise	Table height Z	−12.5 cm
Starting Angle	45	Detector radius	27.5 cm
Image reconstruction method	Filtered back projection	Width & center	20%, 140 keV

**Table 3 diagnostics-13-00611-t003:** Average (±standard deviation (SD)) indices of signal-to-noise (SNR), peak signal-to-noise (PSNR), and contrast-to-noise (CNR) of different filters for the studied myocardial perfusion SPECT images.

CNR ± SD	PSNR (dB) ± SD	SNR ± SD	Filter
2.35 ± 0.5	32.04 ± 3.42	4.34 ± 0.46	Butterworth (*n* = 5, fc = 0.3 Nq)
2.21 ± 0.47	50.09 ± 10.63	4.06 ± 0.40	Gaussian
2.38 ± 0.49	33.67 ± 3.96	4.38 ± 0.45	Wiener (3 × 3)
2.31 ± 0.47	34.17 ± 4.00	4.26 ± 0.45	MMWF (3 × 3)
2.65 ± 0.57	29.93 ± 3.13	4.90 ± 0.69	Wiener (5 × 5)
2.55 ± 0.55	30.29 ± 3.06	4.70 ± 0.63	MMWF (5 × 5)
2.41 ± 0.42	29.51 ± 2.88	4.16 ± 0.68	Hamming (fc = 0.3 Nq)

**Table 4 diagnostics-13-00611-t004:** *p*-values obtained using one way-ANOVA test to compare different filters for the studied myocardial perfusion SPECT images (*p*-values in bold are less than 0.05.).

Index		Filter	Butterworth (*n* = 5, *fc* = 0.3 Nq)	Gaussian	Wiener (3 × 3)	MMWF (3 × 3)	Wiene × (5 × 5)	MMWF (5 × 5)	Hamming (fc = 0.3 Nq)
	Filter								
**SNR**	**Butterworth (*n* = 5, *fc* = 0.3 Nq)**		-	**0.003**	0.666	0.367	**7.8 × ** $10^{-9}$	**1.5 × ** $10^{-4}$	**0.047**
	**Gaussian**		**0.003**	-	**0.001**	**0.034**	**4.9 × ** $10^{-17}$	**4.5 × ** $10^{-11}$	0.30
	**Wiener** **(3 × 3)**		0.666	**0.001**	-	0.183	**7.5 × ** $10^{-8}$	**0.001**	**0.016**
	**MMWF (3 × 3)**		0.367	**0.034**	0.183	-	**4.6 × ** $10^{-11}$	**3 × ** $10^{-6}$	0.277
	**Wiener** **(5 × 5)**		**7.8 × ** $10^{-9}$	**4.9 × ** $10^{-17}$	**7.5 × ** $10^{-8}$	**4.6 × ** $10^{-11}$	**-**	**0.035**	**5.3 × ** $10^{-14}$
	**MMWF** **(5 × 5)**		**1.5 × ** $10^{-4}$	**4.5 × ** $10^{-11}$	**0.001**	**3 × ** $10^{-6}$	**0.035**	**-**	**1.55 × ** $10^{-8}$
	**Hamming (fc = 0.3 Nq)**		**0.047**	0.30	**0.016**	0.277	**5.3 × ** $10^{-14}$	**1.55 × ** $10^{-8}$	**-**
**PSNR**	**Butterworth (*n* = 5, *fc* = 0.3 Nq)**		-	**5.9 × ** $10^{-118}$	**1.7 × ** $10^{-4}$	**1 × ** $10^{-6}$	**2 × ** $10^{-6}$	**6.7 × ** $10^{-5}$	**5.2 × ** $10^{-10}$
	**Gaussian**		**5.9 × ** $10^{-118}$	-	**1.1 ×** $10^{-108}$	**9.8 ×** $10^{-106}$	**5.1 ×** $10^{-129}$	**3.6 ×** $10^{-127}$	**6.1 ×** $10^{-141}$
	**Wiener** **(3 × 3)**		**1.7 × ** $10^{-4}$	**1.1 ×** $10^{-108}$	-	0.249	**3.7 ×** $10^{-16}$	**1 ×** $10^{-13}$	**3.3 ×** $10^{-22}$
	**MMWF** **(3 × 3)**		**1 × ** $10^{-6}$	**9.8 × ** $10^{-106}$	0.249	-	**1 ×** $10^{-19}$	**4.4 ×** $10^{-17}$	**2 ×** $10^{-26}$
	**Wiener** **(5 × 5)**		**2 × ** $10^{-6}$	**5.1 × ** $10^{-129}$	**3.7 ×** $10^{-16}$	**1 ×** $10^{-19}$	-	0.397	0.289
	**MMWF** **(5 × 5)**		**6.7 × ** $10^{-5}$	**3.6 × ** $10^{-127}$	**1 ×** $10^{-13}$	**4.4 ×** $10^{-17}$	0.397	-	**0.047**
	**Hamming (fc = 0.3 Nq)**		**5.2 × ** $10^{-10}$	**6.1 × ** $10^{-141}$	**3.3 ×** $10^{-22}$	**2 ×** $10^{-26}$	0.289	**0.047**	-
**CNR**	**Butterworth (*n* = 5, *fc* = 0.3 Nq)**		-	**0.031**	0.648	0.590	**3 ×** $10^{-6}$	**0.002**	0.329
	**Gaussian**		**0.031**	-	**0.009**	0.106	**2.7 ×** $10^{-11}$	**3 ×** $10^{-7}$	**0.002**
	**Wiener** **(3 × 3)**		0.648	**0.009**	-	0.320	**2 ×** $10^{-5}$	**0.009**	0.604
	**MMWF** **(3 × 3)**		0.590	0.106	0.320	-	**2 ×** $10^{-7}$	**3.4 ×** $10^{-4}$	0.131
	**Wiener** **(5 × 5)**		**3 × ** $10^{-6}$	**2.7 × ** $10^{-11}$	**2 ×** $10^{-5}$	**2 ×** $10^{-7}$	**-**	**0.049**	**1.6 ×** $10^{-4}$
	**MMWF** **(5 × 5)**		**0.002**	**3 × ** $10^{-7}$	**0.009**	**3.4 ×** $10^{-4}$	**0.049**	**-**	**0.036**
	**Hamming (0.3 Nq)**		0.329	**0.002**	0.604	0.131	**1.6 ×** $10^{-4}$	**0.036**	**-**

## Data Availability

The data presented in this study are available on request from the corresponding author.

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
