# Peer review of "Myocardial Perfusion Single-Photon Emission Computed Tomography (SPECT) Image Denoising: A Comparative Study"

_diagnostics, 2023, doi:10.3390/diagnostics13040611_

Round 1

Reviewer 1 Report

The present study aim to evaluate different filters to improve image quality of SPECT myocardial perfusion imaging, using a NaI camera. They report SNR, CNR, PSNR ratios values.

The subject is interesting as myocardial perfusion imaging remains a standard procedure that is commonly used in nuclear medicine units. The impacts remains however limited as there is a growth in the use of CZT cameras and also PET imaging.

A major limit is the lack of visual analysis and comparison between the filtering parameters in terms of visual image quality and final diagnosis. Has the images been blindly reviewed by senior NM physicians? This should be added to the manuscript, with blinded evaluation and rating, and also comparison of diagnosis. Image quality evaluated by SNR, CNR, PSNR is important, but the major fact and the motivation to perform or not the exam is the clinical impact.

Clarify the number of patients : line 103 "30 patients (15 men and 17 women)"; 2 women were excluded because of pregnancy?

Explain the myocardial perfusion protocol: rest/stress in 1 or 2 days? why 15-20mCi at rest and 20-25mCi at stress if 2 days? if 1 day, how to deal with residual activity? was it a weight-based dose?

How were ROI defined ? ie, how was positionned the heart (signal) box, and the noise/background? Figure 4 is not so explicit, was it done by the same operator? automatically?

Did the authors took into account the variability in terms of digestive uptake due to 99mTc tracers? This is a major artifact for myocardial perfusion imaging, and this can lead to variations in SNR independently of the filter used.

BMI of the patients should also be reported.

Author Response

see attached file please

Reviewer 2 Report

I enjoyed reading this paper. Authors demonstrated the difference among different filters to improve the quality of myocardial perfusion SPECT. The approach in this study is sound and appropriate. The data presented in this study is worthwhile and appropriate in the present form.

I believe this study is worthwhile to be published in Diagnostics. At this point, I do not have any comment for the further improvement of this study. 

Reviewer 3 Report

The article “Comparison of Different Denoising Filters in Myocardial Perfusion Single Photon Emission Computed Tomography (SPECT) Images” by Abdurrahim Rahimian and co-author describes the effect of using different denoising filters in SPECT Images. Unfortunately, the result description was not enough. Therefore, the article needs significant modifications.

(1) Please briefly describe the patient’s information involved in this study

(2) In line 32, what is the meaning of ‘leading causes’? Incorporate the statistical standing in recent years.

(3) In line 40, ‘it is not widely used”, How to confirm PET is not widely used? This statement needs correction or current statistical value. However, PET is widely used nowadays.

(4) In Line 73, please incorporate how good the Butterworth filter was (in percentages/ contrast value) than the Hamming filter.

(5) Is it only applicable to myocardial perfusion SPECT images, or could it apply to other SPECTs? If yes, then explain briefly.

(6) What are the prerequisites behind choosing a particular filter for a group of SPECT images? For example, in this study, the wiener filter with a kernel size of 5*5 outperformed the other filters; how can these findings help others reduce the analysis time or choose an optimal filter in the beginning?

(7) Before data acquisition, do these filters influence producing a better image resolution for a specific scanning protocol?

(8) Please incorporate more information about the radiotracer 99?Tc – ??BI involved in this study.

(9) Please incorporate similar statistics presented in Figure 5 for wiener (5*5).

(10) In Figure 6, please include “(x*x represents the kernel size)” for clarity.

(11) The Methods section needs to upgrade with more details about the represented study.

(12) The results must be more elaborate than the present form.

Author Response

Kindly see attached file

Round 2

Reviewer 3 Report

The modified manuscript can be accepted for publication after editorial corrections.